# Berberine Suppresses Leukocyte Adherence by Downregulating CX3CL1 Expression and Shedding and ADAM10 in Lipopolysaccharide-Stimulated Vascular Endothelial Cells

**DOI:** 10.3390/ijms23094801

**Published:** 2022-04-27

**Authors:** Yi-Hong Wu, Chen-Ying Wei, Wei-Chin Hong, Jong-Hwei Su Pang

**Affiliations:** 1Department of Chinese Medicine, Chang Gung Memorial Hospital, Guishan, Taoyuan 333, Taiwan; mzpjih@adm.cgmh.org.tw (Y.-H.W.); hedywei@cgmh.org.tw (C.-Y.W.); cute09272006@gmail.com (W.-C.H.); 2School of Traditional Chinese Medicine, Chang Gung University, Guishan, Taoyuan 333, Taiwan; 3Graduate Institute of Clinical Medical Sciences, Chang Gung University, Guishan, Taoyuan 333, Taiwan; 4Department of Physical Medicine and Rehabilitation, Chang Gung Memorial Hospital, Guishan, Taoyuan 333, Taiwan

**Keywords:** berberine, acute lung injury, lipopolysaccharide, CX3CL1-CX3CR1 axis, soluble fractalkine, ADAM10

## Abstract

Berberine exerts therapeutic effects in inflammation-associated diseases. In a lipopolysaccharide (LPS)-induced endotoxemic acute lung injury (ALI) rat model, berberine alleviated lung injury through different anti-inflammatory mechanisms; however, treatment effects on CX3CL1 expression and shedding remain to be examined. As these processes play important roles in promoting the binding of leukocytes to the endothelium, the CX3CL1/CX3CR1 axis and its related pathways may serve as potential targets for the clinical treatment of ALI. The anti-inflammatory effects of berberine were investigated in LPS-stimulated rats, human umbilical cord vein endothelial cells (HUVECs), and THP-1 monocytic cells. Cx3cl1 expression in rat pulmonary tissues was examined using immunohistochemistry. CX3CL1, CX3CR1, RELA, STAT3, and ADAM10 levels were examined using Western blotting. CX3CL1 and ADAM10 mRNA levels were examined using quantitative real-time polymerase chain reaction. Soluble fractalkine levels in LPS-stimulated rats and HUVECs were examined using the enzyme-linked immunosorbent assay. Berberine significantly mitigated the LPS-induced upregulation of fractalkine and soluble fractalkine in rats and cultured HUVECs. Berberine mitigated the LPS-induced activation of the NF-κB and STAT3 signaling pathways. In THP-1 cells, berberine mitigated the LPS-induced upregulation of CX3CR1. Furthermore, the membrane expression of ADAM10 in LPS-stimulated HUVECs was suppressed by the berberine treatment. Berberine dose-dependently inhibited the LPS-induced activation of the CX3CL1/CX3CR1 axis and fractalkine shedding through ADAM10. These findings reveal a novel molecular mechanism underlying the inhibitory effect of berberine on monocyte adherence to the endothelium during inflammation.

## 1. Introduction

Acute lung injury (ALI) and its severe form, acute respiratory distress syndrome (ARDS), are acute, diffuse, inflammatory conditions. Infections and lung tissue injury promote the release of proinflammatory cytokines [1], which recruit leukocytes to the lungs and damage the capillary endothelium and alveolar epithelium by promoting the release of reactive oxygen species and proteases [2]. The characteristic features of ALI and ARDS are epithelial dysfunction, increased vascular permeability, and excessive endothelial leakage [3]. These pathological changes lead to the accumulation of blood and proteinaceous fluid (edema) in the lung interstitium and the loss of functional surfactants. Ventilation–perfusion mismatch with hypoxemia and decreased lung compliance may lead to clinical complications.

Endothelial dysfunction is characterized by the upregulation of cell adhesion molecules and chemokines, which is a crucial initial step in the recruitment of leukocytes to the foci of inflammation. CX3CL1 (also known as fractalkine), a chemokine with a unique CX3C motif [4], functions as both an adhesion molecule and chemoattractant [5]. CX3CL1 is expressed on the membranes of activated endothelial cells. The N-terminal domain containing the CX3C motif is cleaved and secreted [6]. Tumor necrosis factor-alpha (TNF-α), interleukin (IL)-1β, and lipopolysaccharide (LPS) upregulate CX3CL1 [7]. The chemokine domain of CX3CL1, which is fused to a mucin-like stalk in the transmembrane and cytoplasmic regions to form a cell adhesion receptor, can arrest cells under physiological conditions [8]. In interferon (IFN)-alpha-activated human aortic smooth muscle cells, RELA and STAT1/STAT3 regulate the expression of CX3CL1 by binding to its promoter [9]. Previous studies have reported that the inflammatory chemokine CX3CL1 is associated with the pathogenesis of inflammatory lung diseases [10,11]. The level of soluble CX3CL1 is correlated with disease severity and clinical outcomes in patients with sepsis [12]. Recent studies have reported that the constitutive cleavage of CX3CL1 is predominantly mediated by ADAM10, a membrane protease that regulates monocyte adherence and migration [13].

Berberine has been reported to alleviate LPS-induced lung injury in a mouse model by exerting anti-inflammatory effects [14]. Additionally, berberine inhibits the TNF-α-induced expression of ICAM1 and MCP1 and the activation of NF-κB in human aortic endothelial cells in vitro through the AMPK-dependent pathway [15]. Previously, we demonstrated that berberine inhibited LPS-induced monocyte adherence to the endothelium in vivo and in vitro by downregulating VCAM1 expression [16]. However, the effect of berberine on LPS-induced CX3CL1 expression and shedding in vascular endothelial cells via ADAM10 has not been previously reported on. This study aims to investigate the in vivo and in vitro anti-inflammatory effects of berberine and its regulatory effects on CX3CL1 expression, which may promote the adherence of monocytes to the endothelium, an early inflammatory response induced by LPS.

## 2. Results

### 2.1. Berberine Downregulated Cx3cl1 in the Endothelium of Lung Postcapillary Venules of LPS-Treated Rats

A previous study reported that berberine significantly inhibited the adherence of monocytes to the endothelium in LPS-challenged rat lung postcapillary venules [16]. To examine the regulatory effects of berberine on Cx3cl1 (a well-known cell adhesion molecule involved in the early inflammatory response), rats were pretreated with or without berberine (50 mg/kg body weight) before LPS injection. The expression of Cx3cl1 in rat lung postcapillary venules was examined using an immunohistochemical analysis. As shown in Figure 1, the control group exhibited a mild expression of Cx3cl1 in the partial region of the endothelium of the postcapillary venules. Treatment with LPS for 9 h markedly increased leukocyte adherence to the endothelium of the lung postcapillary venules and upregulated Cx3cl1 expression. Pretreatment with berberine mitigated the LPS-induced upregulation of Cx3cl1 in the endothelium and the number of adherent leukocytes.

### 2.2. Berberine Mitigated the LPS-Induced Upregulation of CX3CL1 in Cultured Human Umbilical Cord Vein Endothelial Cells (HUVECs)

The effect of berberine on the expression of CX3CL1 in LPS-stimulated HUVECs was analyzed using Western blotting. HUVECs were pretreated with 10, 25, or 50 µg/mL berberine for 24 h. The cells were lysed 4 h post-LPS treatment, and the cell lysates were subjected to Western blotting. As shown in Figure 2a, the expression of CX3CL1 was significantly upregulated upon stimulation with 5 µg/mL LPS for 4 h. Pretreatment with 10, 25, or 50 µg/mL berberine dose-dependently downregulated the expression of CX3CL1 in LPS-stimulated HUVECs.

The effect of berberine on CX3CL1 mRNA expression was determined by qRT-PCR using specific primers. As shown in Figure 2b, CX3CL1 mRNA levels were significantly downregulated in HUVECs treated with LPS for 4 h. Pretreatment with 10, 25, or 50 µg/mL berberine for 24 h dose-dependently mitigated the LPS-induced upregulation of CX3CL1 mRNA levels.

### 2.3. Berberine Mitigated the LPS-Induced Activation of NF-κB/STAT3 Pathways in HUVECs

Cultured HUVECs were pretreated with 50 μg/mL berberine for 1 h and then stimulated with 5 μg/mL LPS for 30, 60, or 90 min. The phosphorylated and total levels of RELA, STAT3, and MAPK14 were examined using Western blotting. LPS (5 µg/mL) significantly and time-dependently upregulated the phosphorylation of RELA, STAT3, and MAPK14 (Figure 3). Pretreatment with 50 μg/mL berberine for 1 h significantly mitigated the LPS-induced phosphorylation of RELA and STAT3, but did not affect the LPS-induced phosphorylation of MAPK14.

### 2.4. Berberine Mitigated the LPS-Induced Upregulation of CX3CR1 in THP-1 Cells

The adherence of monocytes to the endothelium is mediated by the binding of CX3CL1 expressed on the endothelial surface to CX3CR1 expressed on the monocyte surface. The effect of berberine on the expression of CX3CR1 in LPS-stimulated THP-1 cells was analyzed using Western blotting. THP-1 cells were pretreated with 10, 25, or 50 µg/mL berberine for 1 h before stimulation with 5 µg/mL lipopolysaccharide (LPS). Cell lysates were prepared at 4 h post-LPS treatment and subjected to Western blotting using anti-CX3CR1 antibodies. As shown in Figure 4, stimulation with 5 µg/mL LPS for 4 h significantly upregulated the expression of CX3CR1 in THP-1 cells. Pretreatment with 10, 25, or 50 µg/mL berberine significantly and dose-dependently mitigated the LPS-induced upregulation of CX3CR1 in THP-1 cells.

### 2.5. Berberine Mitigated the LPS-Induced Upregulation of Soluble Fractalkine In Vivo and In Vitro

The enzyme-linked immunosorbent assay results revealed that the blood levels of soluble fractalkine were significantly upregulated in the LPS group. Rats were intraperitoneally administered berberine at doses of 10, 25, or 50 mg/kg body weight, 1 h before LPS stimulation. As shown in Figure 5a, pretreatment with berberine significantly and dose-dependently mitigated the LPS-induced upregulation of soluble fractalkine levels in the blood of rats. HUVECs were pretreated with 10, 25, or 50 µg/mL berberine for 24 h before stimulation with 5 µg/mL LPS. The soluble fractalkine level in the conditioned medium of cultured HUVECs was significantly upregulated 4 h post-LPS treatment. As shown in Figure 5b, pretreatment with 10, 25, or 50 µg/mL berberine significantly and dose-dependently downregulated the levels of soluble fractalkine in the conditioned media of LPS-stimulated HUVECs.

### 2.6. Berberine Mitigated the LPS-Induced Upregulation of ADAM10 in HUVECs

ADAM10, a protease family member, is involved in shedding CX3CL1 from the cell membrane. Soluble fractalkine functions as a chemokine to attract monocytes to the endothelium. Berberine may mediate the downregulation of soluble fractalkine through ADAM10. The effect of berberine on the expression of ADAM10 in LPS-treated HUVECs was analyzed using Western blotting. As shown in Figure 6, pretreatment with 10, 25, or 50 µg/mL berberine significantly and dose-dependently downregulated the expression of ADAM10 in LPS-stimulated HUVECs.

## 3. Discussion

Currently, therapeutic strategies for ALI and ARDS, which are life-threatening conditions [17], include supportive care, ventilator support, and pharmacological treatment. Several clinical trials have aimed to identify pharmacological agents such as sedative cis-atracurium for early ARDS or low-dose steroids [18]. However, none of these test drugs yielded beneficial outcomes or were used for clinical treatment. Currently, research efforts are focused on identifying novel agents for ALI/ARDS that can target a heterogeneous etiology.

Berberine, an isoquinoline alkaloid isolated from herbal extracts, exerts anti-inflammatory effects [19]. Previous studies have reported that berberine exerts therapeutic effects in the LPS-induced ALI model, which represents the pathological conditions of cytokine release and inflammatory signaling [20] through different regulatory mechanisms. Berberine has been reported to mitigate the LPS-induced TNF-α production and downregulation of cytosolic phospholipase A2 levels, phosphorylation, and thromboxane A2 release [14]. One study reported that berberine alleviated inflammation-associated endoplasmic reticulum stress by modulating the PERK-mediated Nrf2/HO-1 signaling axis [21]. In addition to cytokine regulation, this study examined the mechanism underlying leukocyte adherence to the endothelium during the initial stages of inflammation. Berberine was reported to downregulate VCAM1 expression and inhibit leukocyte–endothelium adhesion [16]. Based on previous findings, this study examined the effects of berberine on other adhesion molecules in the inflammatory endothelium. Accumulating evidence has revealed that the CX3CL1/CX3CR1 axis plays a crucial role in endothelial barrier dysfunction in ALI [3]. The expression of CX3CL1 is a potential biomarker for sepsis [22].

In this study, berberine suppressed Cx3cl1-mediated inflammatory processes by inhibiting leukocyte adherence to the endothelium and by downregulating Cx3cl1 expression in the lungs of LPS-challenged rats. Additionally, berberine mitigated the LPS-induced upregulation of CX3CL1 mRNA and protein levels by suppressing the phosphorylation of RELA and STAT3, which are upstream transcription factors involved in activating the transcription of *CX3CL1* [9]. Berberine mitigated the LPS-induced upregulation of CX3CR1 in THP-1 cells. These findings demonstrated that berberine could inhibit the CX3CL1/CX3CR1 axis in a dose-dependent manner and, consequently, suppress the adherence of monocytes to the endothelium.

Furthermore, berberine mitigated the LPS-induced upregulation of serum Cx3cl1 in rats. Berberine also mitigated the LPS-induced upregulation of ADAM10 in HUVECs. This indicated that berberine suppressed the levels of soluble fractalkine by downregulating ADAM10 expression. ADAM10, a member of the protease family that is widely expressed in several tissues [23], promoted the cleavage of CX3CL1. Cleaved CX3CL1 plays a crucial role in regulating cell–cell adhesion, leukocyte migration, and alveolar leukocyte recruitment [24]. Several endogenous mediators subjected to proteolytic cleavage by ADAM family members are involved in the pathogenesis of acute and chronic pulmonary inflammatory disorders [25]. ADAM family members have been reported to be potential therapeutic targets for inflammatory lung diseases, as they are involved in the shedding of TNF-α, adhesion molecules, or proteoglycans [26]. The cleavage of the membrane-bound isoform of the receptor for advanced glycation end products (RAGE) and the upregulation of soluble cleaved RAGE levels in the alveolar space sustain inflammation under these conditions [27]. In addition to exerting anti-inflammatory effects, berberine can downregulate LPS-induced ADAM10 expression and suppress the levels of inflammatory cleaved products. Thus, berberine is a potent anti-inflammatory drug that alleviates lung inflammation associated with ADAM10-mediated lung diseases.

## 4. Materials and Methods

### 4.1. Drugs and Reagent

Berberine chloride, ethanol (32221), dimethyl sulphoxide (DMSO)(D2650), and LPS were purchased from Sigma-Aldrich (St. Louis, MO, USA). Anti-CX3CL1 (ab25088), anti-CX3CR1 (ab8021), and anti-ADAM10 (ab1997) antibodies were obtained from Abcam (Cambridge, UK). Anti-phospho-STAT3 (D3A7), anti-RELA (D14E12), and anti-phospho-MAPK14 antibodies were purchased from Cell Signaling Technology (Danvers, MA, USA).

Berberine chloride was dissolved in H_2_O:ethanol (1:1) for in vivo experiments and in DMSO for in vitro experiments performed in this study.

### 4.2. Animals

All animal experiments were approved by the Animal Experiment Committee of Chang Gung University and performed according to the guidelines for the care and use of laboratory animals. Male Sprague Dawley rats with a body weight of 200–250 g were purchased from BioLasco Taiwan Co. and maintained on a normal rat chow diet for a week before the experiments.

### 4.3. Animal Experiments

Rats were randomly divided into the following five groups: control group, intraperitoneally administered with isotonic sodium chloride solution (0.2 mL/10 g bodyweight); LPS group, intraperitoneally administered with LPS (5 mg/kg bodyweight; 0.2 mL/10 g bodyweight); berberine 10 group, administered with berberine (0.1 mL/10 g bodyweight; 10 mg/kg bodyweight), followed by administration of LPS at 1 h post-berberine treatment; berberine 20 group, administered with berberine (0.1 mL/10 g bodyweight; 20 mg/kg bodyweight), followed by administration of LPS at 1 h post-berberine treatment; berberine 50 group, administered with berberine (0.1 mL/10 g bodyweight; 50 mg/kg bodyweight), followed by administration of LPS at 1 h post-berberine treatment. The rats were sacrificed 9 h after the LPS treatment. Whole blood samples were collected and serum was prepared for the enzyme-linked immunosorbent assay. Lung tissues were excised, fixed with 10% buffered formalin, and embedded in paraffin. Paraffin-embedded tissues were sectioned to a thickness of 4 µm. The expression of Cx3cl1 in pulmonary sections was examined using immunohistochemical analysis.

For immunohistochemical analysis, the paraffin-embedded pulmonary sections were incubated at 60 °C, deparaffinized in xylene, and rehydrated in a series of gradually decreasing concentrations of ethanol. The sections were washed with 1× phosphate-buffered saline (PBS) and incubated with antigen retrieval buffer (100 mM Tris, 5% (*w*/*v*) urea, pH 9.5) at 95 °C for 10 min to retrieve the antigens. Next, the sections were washed with 1× PBS, incubated with 3% H_2_O_2_ for 10 min to inhibit endogenous peroxidase activity, and blocked with blocking solution (1% bovine serum albumin and 1% goat serum) for 30 min. The sections were then incubated with the primary antibodies in blocking solution for 2 h, washed, and incubated with peroxidase-conjugated secondary antibodies in 1× PBS for 1 h. Immunoreactive signals were developed using 3,3-diaminobenzidine. Nuclei were counterstained with hematoxylin. The cells were examined under a microscope.

### 4.4. Cell Culture

HUVECs were purchased from the Bioresource Collection and Research Center (Taiwan, China). Cells were cultured to confluence in Medium 199 supplemented with 20% fetal bovine serum (FBS) at 37 °C in a humidified 95% air/5% CO_2_ incubator. HUVECs passaged 3–5 times were used for the experiments. THP-1 cells (human monocytic leukemia cells) were obtained from the American Type Culture Collection and cultured in Roswell Park Memorial Institute-1640 medium supplemented with 10% FBS and antibiotics at 37 °C in a humidified atmosphere containing 5% CO_2_/95% air.

### 4.5. MTT (3-[4, 5-Dimethylthiazol-2-yl]-2, 5-Diphenyltetrazolium Bromide) Assay

Cells treated with or without berberine for 24 h were washed once with PBS and incubated in 1 mL of Medium 199 containing 0.05 mg/mL of MTT at 37 °C for 1 h. The medium was removed and the formazan crystals in the cells were solubilized in 1 mL of dimethyl sulfoxide. The optical density of the mixture was measured at 570 nm wavelength using a spectrophotometer.

### 4.6. RNA Isolation

To isolate total RNA, cells were lysed in guanidinium isothiocyanate buffer and subjected to one-step phenol–chloroform–isoamyl alcohol extraction. Briefly, 5 × 10^6^ cells were lysed in 0.5 mL solution D containing 4 M guanidium isothiocyanate, 25 mM sodium citrate (pH 7.0), 0.5% sodium sarcosine, and 0.1 M β-mercaptoethanol with vigorous vortexing. Next, 50 µL of 2 M sodium acetate (pH 4.0), phenol (0.5 mL), and chloroform-isoamyl alcohol (100 µL, 49:1, *v*/*v*) were sequentially added to the homogenate. The mixture was vortexed for 30 s and centrifuged at 10,000× *g* at 4 °C for 15 min. To precipitate RNA, the mixture was incubated with isopropanol (0.5 mL) at −20 °C for 1 h. The sample was centrifuged at 10,000× *g* at 4 °C for 15 min. The RNA pellet was rinsed with ice-cold 75% ethanol, dried, and dissolved in diethyl pyrocarbonate-treated double distilled water.

### 4.7. Quantitative Real-Time Polymerase Chain Reaction (qRT-PCR)

Total RNA (1 µg) was reverse transcribed into complementary DNA (cDNA). Reverse transcription was performed in a 20 µL reaction mixture containing 200 U of reverse transcriptase, 0.25 µg of random primers, and 0.8 mM dNTPs at 42 °C for 1 h. qRT-PCR analysis was performed using 2 µL of cDNA as a template. *GAPDH* was used as an internal control. PCR was performed in a buffer containing 10 mM Tris-HCl (pH 8.3), 50 mM KCl, 1.5 mM MgCl_2_, 0.2 mM dNTPs, 1 µM of each primer, and 5 U Taq DNA polymerase. PCR conditions were as follows: 30 cycles of denaturation at 94 °C for 1 min, annealing at 55 °C for 1 min, and extension at 72 °C for 2 min. The specific primer sequences used in the qRT-PCR analysis were as follows: *CX3CL1*, 5′-GAGCC GACTC CTTCT TCC-3′ (sense) and 5′-CCTCC ATCCT GAGCC TTT-3′ (antisense); *GAPDH*, 5′-TTCAT TGACC TCAAC TACAT-3 (sense) and 5-GAGGG GCCAT CCACA GTCTT-3′ (antisense).

### 4.8. Western Blotting Analysis

Cells were sonicated in lysis buffer (Tris-HCl (pH 7.5), 150 mM NaCl, 1 mM ethylenediaminetetraacetic acid, 2 mM dithiothreitol, 2 mM phenylmethylsulfonyl fluoride, and 1% Triton X-100). The protein concentration in the lysate was determined using the Bradford assay (Bio-Rad Laboratories, CA, USA). Equal amounts of protein were subjected to sodium dodecyl sulfate-polyacrylamide gel electrophoresis on a 12% gel. The resolved proteins were transferred onto polyvinylidene difluoride membranes. The membrane was blocked with blocking solution (1% bovine serum albumin and 1% goat serum in PBS) for 1 h at room temperature. The membrane was then incubated with primary antibodies in a blocking solution for 2 h. After washing three times with PBS, the membrane was incubated with goat horseradish peroxidase-conjugated antimouse IgG secondary antibodies in PBS for 1 h. The membrane was washed three times with PBS, and immunoreactive signals were detected using an enhanced chemiluminescence system (Amersham Pharmacia Biotech, Little Chalfont Buckinghamshire, UK).

### 4.9. Statistical Analysis

Data are expressed as mean ± standard error of the mean. Means were compared using independent and paired Student’s *t*-tests for unpaired and paired samples, respectively. Means between multiple groups were compared using a one-way analysis of variance.

## 5. Conclusions

This study elucidated a novel molecular mechanism underlying the anti-inflammatory effects of berberine. In particular, this study demonstrated that berberine inhibits monocyte adherence to LPS-stimulated HUVECs. Berberine effectively inhibited the activation of the CX3CL1/CX3CR1 axis by downregulating CX3CL1 mRNA and protein levels in LPS-induced HUVECs by inhibiting the NF-κB and STAT3 pathways and downregulated the expression of CX3CR1 in monocytes. Additionally, berberine downregulated the levels of ADAM metallopeptidase domain 10 and soluble fractalkine both in vivo and in vitro. These results suggest that berberine is a potential therapeutic agent for ALI-associated endothelial dysfunction and that it targets the CX3CL1–CX3CR1 axis. In addition to the previously reported mechanisms of berberine, future studies must focus on the therapeutic applications of berberine in other diseases that exhibit inflammatory events similar to those in ALI.

## Figures and Tables

**Figure 1 ijms-23-04801-f001:**
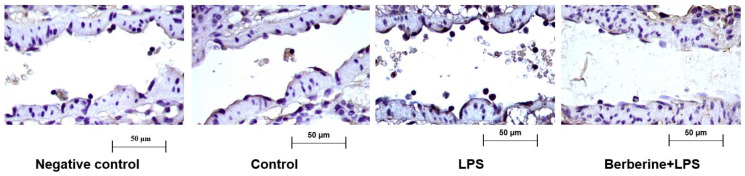
Berberine mitigated the lipopolysaccharide (LPS)-induced upregulation of Cx3cl1 in the endothelium of rat lung postcapillary venules. Rat lung tissues were subjected to immunohistochemical analysis to examine the expression of Cx3cl1. Negative control samples were processed using the same protocol, except for the use of anti-Cx3cl1 primary antibodies. Brown color in the endothelium indicates the expression of Cx3cl1. Four microphotographs representing the pulmonary section of the negative control, control, LPS (5 mg/kg bodyweight)-treated, and berberine (50 mg/kg bodyweight, intraperitoneal; 1 h)-treated/LPS (5 mg/kg bodyweight, intraperitoneal)-treated groups were shown.

**Figure 2 ijms-23-04801-f002:**
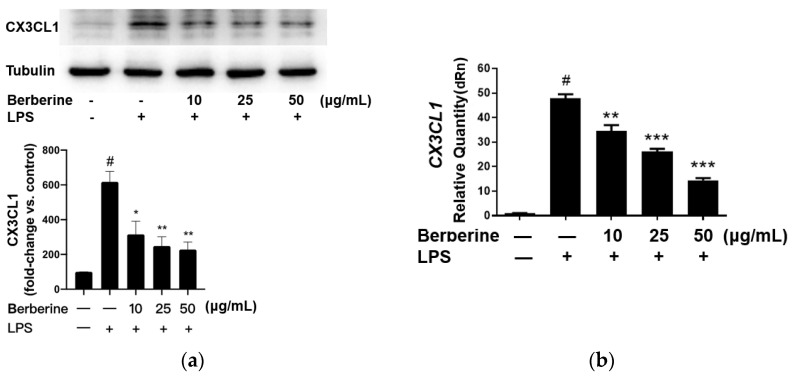
Berberine mitigated the lipopolysaccharide (LPS)-induced upregulation of mRNA and protein levels of CX3CL1 in cultured human umbilical cord vein endothelial cells. (**a**) Cells were pretreated with or without berberine at different concentrations as indicated for 24 h, followed by treatment with LPS for 4 h. Western blotting analysis was performed to examine the levels of CX3CL1. Densitometric analysis data are shown in the bar graph (mean ± standard error of mean). (**b**) Cells were treated as in (**a**). Total RNA was isolated for quantitative real-time polymerase chain reaction analysis. *GAPDH* was used as an internal control. ^#^
*p* < 0.001, compared with the control group; * *p* < 0.05 compared with the LPS group; ** *p* < 0.01 compared with the LPS group; *** *p* < 0.001 compared with the LPS group.

**Figure 3 ijms-23-04801-f003:**
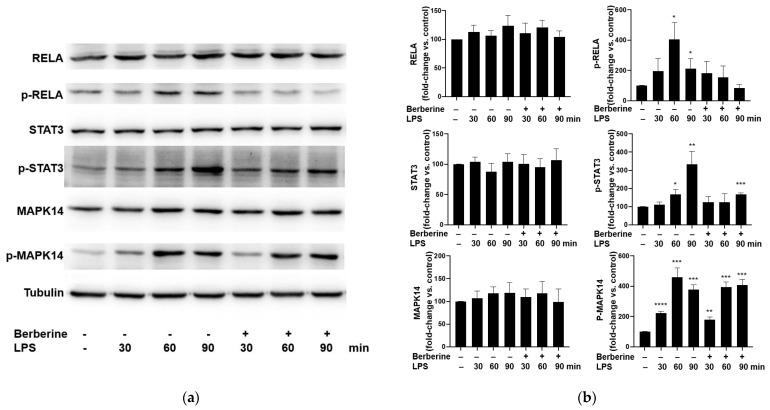
Berberine mitigated the lipopolysaccharide (LPS)-induced phosphorylation of RELA and STAT3 in human umbilical vein endothelial cells. (**a**) The cultured cells were pretreated with or without 50 µg/mL berberine for 1 h, followed by treatment with 5 µg/mL LPS for 30, 60, or 90 min. The levels of RELA, phospho-RELA, STAT3, phospho-STAT3, MAPK14, and phospho-MAPK14 were examined using Western blotting. Tubulin was used as an internal control. (**b**) Densitometric analysis data are represented as mean ± standard error of mean. * *p* < 0.05 compared with the control group; ** *p* < 0.01 compared with the control group; *** *p* < 0.001 compared with the control group; **** *p* < 0.0001 compared with the control group.

**Figure 4 ijms-23-04801-f004:**
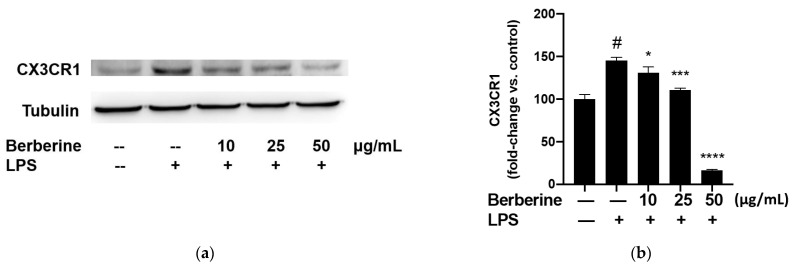
Berberine dose-dependently mitigated the lipopolysaccharide (LPS)-induced upregulation of CX3CR1 in THP-1 cells. (**a**) THP-1 cells were pretreated with 10, 25, or 50 µg/mL berberine for 1 h, followed by stimulation with 5 µg/mL LPS for 4 h. Cell lysates were prepared and subjected to Western blotting analysis. (**b**) Densitometric analysis results are expressed as mean ± standard error of mean. ^#^
*p* < 0.001 compared with the control group; * *p* < 0.05 compared with the LPS group; *** *p* < 0.001 compared with the LPS group; **** *p* < 0.0001 compared with the LPS group.

**Figure 5 ijms-23-04801-f005:**
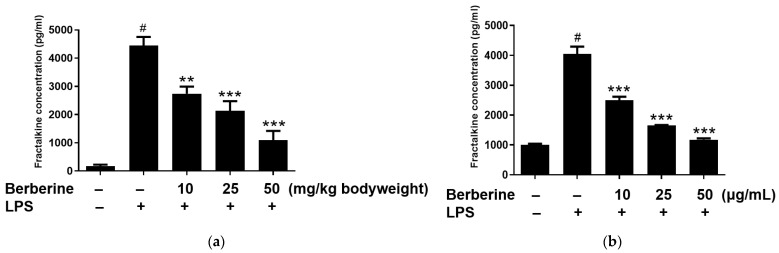
Berberine dose-dependently mitigated the lipopolysaccharide (LPS)-induced upregulation of soluble fractalkine in vivo and in vitro. (**a**) Rats were intraperitoneally pretreated with berberine at doses of 10, 25, or 50 mg/kg bodyweight for 1 h, followed by administration with LPS (5 mg/kg body weight). The serum levels of soluble fractalkine were examined using enzyme-linked immunosorbent assay (ELISA) and the results are represented in the bar graph (mean ± standard error of mean (SEM)). (**b**) Human umbilical vein endothelial cells were pretreated with 10, 25, or 50 µg/mL berberine, followed by stimulation with 5 µg/mL LPS for 4 h. Soluble fractalkine concentration in the conditioned medium was measured using ELISA and the results are represented in the bar graph (mean ± SEM). ^#^
*p* < 0.001 compared with the control group; ** *p* < 0.01 compared with the LPS group; *** *p* < 0.001 compared with the LPS group.

**Figure 6 ijms-23-04801-f006:**
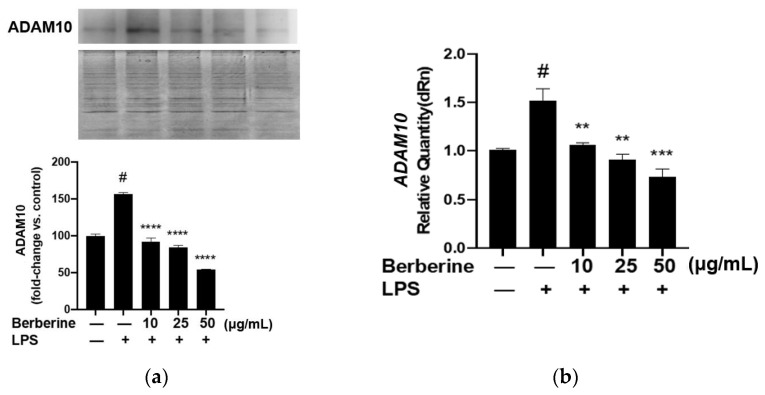
Berberine mitigated the lipopolysaccharide (LPS)-induced upregulation of membrane ADAM10 mRNA and protein expression levels in human umbilical vein endothelial cells. (**a**) Cells were pretreated with 10, 25, or 50 µg/mL berberine for 1 h, followed by stimulation with 5 µg/mL LPS for 4 h. The membrane expression of ADAM10 was analyzed using Western blotting (upper panel). Coomassie blue staining revealed that the protein amounts were equal in different samples (lower panel). Densitometric analysis data are shown in the bar graph (mean ± standard error of mean). (**b**) Cells were treated as in (**a**). Total RNA was isolated and subjected to quantitative real-time polymerase chain reaction analysis. *GAPDH* was used as an internal control. ^#^
*p* < 0.001 compared with the control group; ** *p* < 0.01 compared with the LPS group; *** *p* < 0.001 compared with the LPS group; **** *p* < 0.0001 compared with the LPS group.

## Data Availability

All the data used to support the findings of this study are presented in this article.

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
