# Peer review of "Berberine Suppresses Leukocyte Adherence by Downregulating CX3CL1 Expression and Shedding and ADAM10 in Lipopolysaccharide-Stimulated Vascular Endothelial Cells"

_ijms, 2022, doi:10.3390/ijms23094801_

Round 1

Reviewer 1 Report

The manuscript describes the identification of one of the mechanisms of the anti-inflammatory action of berberine. The in vitro and in vivo experiments were carried out correctly, the discussion corresponds to the results obtained, and the conclusions are beyond doubt. However, the article contains a small number of shortcomings that should be corrected.

1) it is necessary to give the procedure for preparing solutions (formulations) of berberine both for animal experiments and for vitral experiments. Berberine is practically insoluble in isotonic solution.

2) line 81 indicates the dose of berberine 50 mg/kg, referring to fig. 1, in the caption to Fig. 1 (line 96) - dose of berberine 10 mg/kg.

3) line 96-97 - incomprehensible text

Author Response

1) Berberine chloride was dissolved in H2O : ethanol (1:1) for in vivo experiments and in DMSO for in vitro experiments performed in this study. (line 264-265)

2) Dose of berberine in the caption to Fig. 1 (line 96) was corrected as “50 mg/kg”.

3) Incomprehensible text in line 96-97 was rephrased to complete the meaning of the sentence.

Reviewer 2 Report

The manuscript written by Yi-Hong Wu and co-authors is interesting and demonstrates the molecular mechanism underlying the anti-inflammatory effects of Berberine in LPS-induced acute lung injury. The results are presented very clearly, and conclusions correspond to the data. The demonstrated mechanism is novel and would be interesting for researchers. The manuscript could be accepted in its present form.

Author Response

Thank you so much for the positive comments on our manuscript.